# A Complementary View to Computational Thinking and Its Interplay with Systems Thinking

**Ali Hamidi** [1,*] **, Anita Mirijamdotter** [1] **and Marcelo Milrad** [2]

1   Department of Informatics, Faculty of Technology, Linnaeus University, 352 52 Växjö, Sweden
2   Department of Computer Science and Media Technology, Faculty of Technology, Linnaeus University, 352 52 Växjö, Sweden
*   Correspondence: ali.hamidi@lnu.se

**Abstract:** Computational Thinking (CT) pervasively shares its methods, practices, and dispositions across other disciplines as a new way of thinking about problem-solving. Few studies have been carried out studying CT from an Information Systems (IS) perspective. This study elaborates on how systems thinking (ST), an acknowledged theory in the IS field, bonds to CT to address some well-known common issues related to CT such as reductionism and dogmatism, and to supplement the computing nature of CT with behavioral and societal facets involved in its implications. We studied how ST is applied to CT research in the literature. To do so, two primary approaches have been identified that link ST and CT. First, ST is embedded in CT practices meaning that ST is considered as a component of CT. Second, ST and CT are parallelly studied, and ST is considered as a supplementary concept to CT. Correspondingly, we propose a complementary approach that looks at CT from the ST lenses to provide a clearer picture of CT in an educational context. Moreover, we expect this new perspective can help to broaden the development of educational CT concepts and scenarios by including new notions such as framework, interpretation, norms, paradigm, and context.

**Keywords:** computational thinking; systems thinking; information systems; K-12 education





## 1. Introduction

The recent and very rapid increase related to the use of digital technologies in everyday life activities makes it essential to improve people's digital competence and computational thinking (CT) skills, particularly K-12 students who are soon entering the career market in the computational world [1]. Thus, CT is determined as an essential skill for youth development in the 21st century [2]. Considering that CT has a long historical timeline inherited and achieved from many fields [3], it is neither limited only to computing nor to how computer scientists think. Research in CT has been pursued by scholars in computer science (CS), and other disciplines such as biology, history, geography, pedagogy, and statistics [4]. According to Shute et al. [5], CT brings a perspective on human and computer interaction to solve structured along with non-structured problems. Grover and Pea [6] argued that CT has the potential to be introduced and used in all disciplines for the sake of creative and innovative problem-solving. However, Information Systems (IS) discipline still lacks the research for studying CT in its domain [7]. Particularly, our scoping review of the literature in prestigious IS journals (such as basket of eight) reveals no evidence of publications that specifically study CT.

Studying digital competence and CT through the lenses of the IS discipline provides the opportunity to benefit from IS theories and concepts, considering that IS is a multi-disciplinary field dealing with different strategies and operational activities where people, information, and digital technology co-exist and interact with each other [8]. The IS field has the potential to frame its propositions surrounding CT using theories, methods, concepts, and strategies that are different, but still complementary to those applied in CS or pedagogy

such as design theory and socio-technical concepts. In IS, the focus is on the application of any kind of technology rather than the technology itself to consider the human and technology interaction [9]. Disciplinary questions in the IS domain include structural (ontological), epistemological, and socio-political aspects [10]. Therefore, such questions have the capability to cover many dimensions of CT development that are still less studied. Moreover, to develop an IS discourse, the process of theorizing from other disciplines and synthesizing them for contributing to IS field should be taken into consideration [11]. According to Denning and Tedre [3] (p. 213), 'CT is a welcome addition to other fields', so it could open up new possibilities in IS field. Mørch and Kafai [12] argued that CT belongs to disciplines and intellectual fields that can be enriched by computation, which is in line with IS-related research in terms of digitalization and computation.

When it comes to CT in K-12 education, the research and practice on developing CT in schools and bringing it into science, technology, engineering, and mathematics (STEM) subjects have been actively pursued within academic studies. Despite remarkable worldwide attempts to bridge CT into K-12 education, there are still vigorous theoretical and practical debates and issues around the topic [1]. Having said that, looking at the notion of CT from an IS view would be beneficial from two standpoints. First, concepts, theories, and methods from IS discipline can be applied to CT in terms of conceptualization and implementation, which consequently helps to unfreeze taken-for-granted understandings of CT. Second, it could help to decrease the flaws associated with CT that will be described in the next sections such as reductionism and dogmatism tendencies.

Thus, the purpose of this study is to look at CT from the IS lens. To do so, we select *systems thinking* (ST) as an acknowledged theory in the IS field and study how CT and ST are connected as the basis of our paper, bearing in mind that bonding ST to CT is relatively new in the existing body of CT research [12–15]. We argue that this paper tackles an emerging topic that would benefit from uncovering potential theoretical grounds [16]. More concretely, the purpose of this study is to address a research gap about a potential link between CT and ST that has rarely been studied in the literature. From a wider perspective, this paper aims to integrate CT and ST, which have frequently been studied separately in the two fields of CS and IS, and to analyze and characterize CT by addressing its relationships and boundaries in IS research focusing on ST. Therefore, this study seeks to answer the following questions:

In which ways does computational thinking benefit from systems thinking core concepts and theories?

How is systems thinking being introduced into computational thinking research?

What are the possible approaches for applying systems thinking to computational thinking?

To do so, we conducted a scoping literature review and looked at the current state of the link between CT and ST. This paper starts by providing an overview of the underlying concepts of this study including CT and ST. We continue with identifying central problems in CT by employing an IS perspective. Thereafter, we elaborate on the interplay of ST and CT to be followed by a discussion section on classifying the approaches of applying ST to CT. Lastly, conclusions are presented and future research is discussed.

## 2. Revisiting Computational Thinking

Historically, CT was identified as algorithmic thinking in the 1950s [17]. The origin of CT in education is commonly traced to the student-centered works of Seymour Papert in 1980, based on the constructionist approach [18]. A fresh perspective was presented by Jeannette Wing's prominent paper in 2006, which introduced CT as a universally applicable skill and a new kind of literacy for everyone [19]. She introduced it parallel to reading, writing, and arithmetic (known as the three R's) as a formative skill. Subsequently, global attention has been given to CT and education on learning and teaching within STEM subjects. Looking at CT in the literature shows that not only is there a clear-cut definition, but also the term is somehow mystifying over the two last decades [5]. Among different

definitions provided for CT, Aho [20] defined it as a thought process in problem-solving that can be effectively carried out by a computer (either human or machine). In addition to the generic definitions of CT, definitions from other perspectives have also been provided which are built on CT's main elements and CT operational aspects. CT definition based on its core elements focuses on key CT competencies rather than the definition [21]. According to Wing's view [19], the core concepts of CT are abstractions and layers, and the relationship between them. Abstractions are mental tools that are needed for problem-solving, and layers are referred to as different levels involved in the problem-solving process. These two concepts together with other concepts like algorithmic thinking and automation are fundamental to CT [4]. Denning and Tedre [3] described it as a mental skill and practice for designing computations and for explaining the information processes. Two aspects of designing and explaining reflect the engineering tradition and science tradition of computing, respectively.

From an operational perspective, there is a big body of research on the application of CT in classrooms to build and develop CT skills across different subject areas in the curriculum. Accordingly, different approaches have been used including the use of maker technologies and educational robotics [22,23]. A CT framework has been developed through design-based learning activities by Brennan and Resnick [24], where students developed their CT skills by designing games using the Scratch programming language. The framework consists of three key dimensions of CT including concepts, practices, and perspectives. CT concepts are a collection of computational concepts that are used in programming languages. CT practices are those concerned with the learning and thinking processes, and the CT perspectives are reflecting social aspects of CT that are not captured by concepts and practices. According to this framework, CT concepts include: sequences (an order of instructions), loops (operating the same sequences multiple times), events (cause and effect relationships), parallelism (running parallel sequences together at the same time), conditionals (decision-making in certain conditions), operators (performing numeric and string operation), and data (storing, retrieving, and updating values). CT practices are outlined as: being incremental and iterative (the adaptive process in approaching the solution), testing and debugging (development through trial and error), reusing and remixing (building on others' work), abstraction and modularization (building abstract level for generalization). CT perspectives have three elements: expressing (using computation for design and self-expression); connecting (enriching social practices and interaction); and questioning (abilities to negotiate the realities of the technological world).

In 2018, Grover and Pea merged the CT perspectives emerging from Brennan and Resnick's framework into CT practices and represented a comprehensive framework emphasizing problem formulation as a significant part of CT [6]. As for the concepts, CT encompasses the following elements: logic and logical thinking, algorithm and algorithmic thinking, pattern and pattern recognition, abstraction and generalization, evaluation, and automation. CT practices are approaches that are followed in computational problem-solving. These include: problem decomposition, creating computational artifacts, testing and debugging, incremental development, and collaboration and creativity [6]. A brief description of CT concepts and practices indicated by Grover and Pea [6] is presented in Table 1.

It is worth mentioning that CT concepts and practices abovementioned are the public faces of CT in K-12 education. However, some experts contend that CT might be used in other fields that rely heavily on computation, such as cloud, virtual reality, software engineering, data analytics, and artificial intelligence [25]. CT definitions are not limited only to typical human–computer interaction concepts. Mishra et al. [26] argued that CT fosters human creativity and develops new forms of expression. In addition, a number of dispositions and predispositions are shaped in CT complex problem-solving including confidence, perseverance, tolerating ambiguity, the capacity to handle open-ended problems, communication skills, and the ability to collaborate with others in pursuit of a shared objective or solution [27]. Although CT has been referred to as a very promising problem-

solving skill in the educational context, challenges and issues are raised simultaneously in conceptual, empirical, pedagogical, and assessment-related aspects of CT [28]. It shifts the focus of studies to add complementary views to CT, enabling it to compensate for its insufficiencies and to frame future research directions. One of those views is systems thinking, which is defined and described in the Section 3.

**Table 1.** CT key elements [6].

| Dimension | Component | Description |
|---|---|---|
| CT concepts | Logic and Logical thinking | Analyzing a condition to make a decision |
| | Algorithm and algorithmic thinking | Step-by-step plan/procedure to solve a problem |
| | Patterns and pattern recognition | Recognizing a repeating pattern, iteration or recursion in the problem-solving process |
| | Abstraction and Generalization | Simplifying and managing complexity through information hiding (the act of black-boxing), and applying solutions to similar problems |
| | Evaluation | Evaluation of solutions based on efficiency constraints |
| | Automation | An automated solution that will be efficiently executed by a machine |
| CT Practices | Problem decomposition | Breaking a problem down into smaller sub-problems to make it tractable and manageable |
| | Creating computational artifact | Creating computational model, simulation, or prototype that could be used by others |
| | Testing and debugging | Detecting errors in a solution and trying to fix them |
| | Incremental development | Improving the solution through iterative refinement |
| | Collaboration and creativity | Critical competencies that are shaping peer computational practices |

### 3. Defining Systems Thinking

ST is a body of theory, method, and practice that frames the understanding of phenomena and facilitates potential future action [29]. Hoverstadt [30] sketched out nine central concepts of ST including: emergence, holism, modeling, boundaries, difference, relating, loops, complexity, and uncertainty. While emergence is the central property of a system that is not predicted from the behaviors of its parts, other abovementioned concepts of ST are part of and depend on it. In other words, drawing on the emergent properties that stem from the interaction of different components of a system, the whole is consequently greater than a simple addition of its elements' properties. Table 2 provides a brief definition of each concept. Among different perspectives on ST, the three most widely recognized approaches are referred to as the hard, soft, and critical traditions. Each tradition has its assumptions, approaches, and methods. For example, while in hard systems, the focus is on classical cybernetics, systems engineering, and systems dynamics, the soft ST looks at inquiring systems design, soft systems methodology, and cognitive mapping. Critical ST addresses inadequacies of hard and soft approaches through systemic intervention [31]. Regardless of the type of tradition presented above, ST is a driving approach for studying complex real-world problems. There are four common rules in ST approaches known as DSRP, which make it applicable for tackling problems. These fundamental rules include: making a distinction between an identity and another (D); organizing systems that consist of part and whole (S); identifying relationships between identities (R); and taking perspectives (P) [32]. These rules simultaneously drive and are driven by boundary judgment and boundary critique, which are considered as the first step for a systemic intervention for improving a problematic situation [33]. Boundary judgment at the core of ST reflects on identifying the primary (core) and secondary (marginal) boundaries that a phenomenon is being studied. Considering that ST is a holistic approach for resolving complex issues [31], it complements other conventional thinking skills, including CT, making them more productive to engage with and improving situations of real-world complexity.

**Table 2.** ST key concepts [30].

| Concept | Description |
| --- | --- |
| Emergence | Property of a system that cannot be reduced to the sum of property of its elements |
| Holism | A thought process at the level of the system rather than the parts |
| Modeling | Mapping out the pattern of the emergent property of a system |
| Boundaries | The identity of a system that deliberately defines what is inside and what is outside of it |
| Difference | The differentiation of a system from its surrounding environment |
| Relating | Internal and external connection of system, sub-system, and its elements |
| Dynamics and loops | Sets of feedback loops and circular relationships contradictory to linear cause and effect |
| Complexity | Dynamic possible states of a system driving from structure and function |
| Uncertainty | Inevitable ambiguity of emergent property of a system and its perception |

## 4. Problematizing of CT with an ST Lens

Problematizing CT sheds light on unseen aspects of it, both conceptually and operationally, by challenging the underlying assumptions and theories that have been in use. In other words, considering that CT originates from the CS discipline some issues would be questioned that might not be asked by CS scholars. Examples are the kind of IT artifact that is most suitable for CT development, whether different environments require different digital tools for CT practices, and if different approaches and strategies for CT development can be integrated. In a broader view, problematizing CT could be related to human–computer interaction, global challenges, and sustainability aspects. In ST language, it is reflected as pushing out the boundaries of a system and setting out a wider boundary [33]. Recently, scholars raised the issues around CT. They consider that CT claims are overreaching and will not meet its promising expectations [34]. The question is whether CT is considered as a general problem-solving skill or applicable only to CS-oriented issues [1]. Elaborating on critics to the CT definition, Martins-Pacheco et al. [2] referred to some aspects and concepts that do not belong solely to CT. In addition, some scholars argue that CT is a conceptual toolbox only [13,14,28]. They argued that the CT for problem-solving is intrinsically reductionist and ignores the multiple perspectives associated with the problem situation. Reductionism is a point of view implying that an issue can be deconstructed into its constituent parts with casual and straightforward connections [29], which encourages surface-level thinking that is inappropriate when dealing with complicated systems in the real world [30].

Moreover, since the commencement introduction of CT in 2006 as a resonance, scholars have raised critiques and issues about it [2,12–14,28,34]. In 2014, Easterbrook evidently introduced the application of ST to CT in his article entitled "From Computational Thinking to Systems Thinking". The author underlined the necessity of supplementing CT with ST methods for the sake of sustainability [13]. According to him, CT has a major limiting factor that underestimates the social and environmental impacts of the technology. The multiple perspectives and timescales are the factors that would be considered when systems approaches are applied. There are three central gaps in CT approaches that would be improved through ST. The first matter is the domain ontology for sustainability thinking that is missing from the CT view. The second failing of CT is related to the lack of a conceptual tool for reasoning about the notion of change in a complex system. The third limitation of CT is the lack of encouraging critical thinking apparent in CT studies, where the priority is on the immediate functionality of technology rather than the impact on the larger context [13].

Other significant risks looming over CT include the lack of ambition, dogmatism, a narrow view of computing, and the risk of overemphasizing the formulation. Lack of ambition is due to focusing on computing knowledge only. Dogmatism means that CT is the one and the best way of thinking for problem-solving. The narrow view of computing is

derived by limiting CT to programming practices only. Overemphasizing the formulation leads to the risk that all problems can be solved by formulation while ignoring the role of designing aspects [35].

On this note, there is a need to move beyond CT and to think out-of-the-box to solve the problems that do not have computational solutions, such as social challenges, wicked problems, ethical dilemmas, and usefulness judgments [13]. Therefore, scholars attempted to consider and study the social aspects of CT by reflecting on a shift toward computational literacies in a sociocultural framework to support cognition and communication [1]. These social features are the foundation for the CT perspectives that are a core dimension in Brennan and Resnick's framework, for instance [24]. Here, the theories from other disciplines, like IS, would be applicable to achieve a broader awareness of the questions around CT and to identify the dispositions and predispositions involved in CT practices [27]. ST is a commonly traced theory in IS discipline that looks promising in tackling the issues associated with CT, such as the reductionism nature of CT and the trap of dogmatism that are common risks standing over CT practices [35]. Reynolds and Holwell [31], pointed out that the two mentioned traps are consequences of non-systems thinking. According to them, where reductionism lies in overlooking the interconnectivity between variables of a phenomenon, dogmatism rests on a single unconditional perspective.

Drawing on the CT definition from its facets perspective described in Section 2 and the central concept of holism in ST presented by Hoverstadt [30], Figure 1 shows the sequences of thinking in problem-solving based on ST and CT approaches. Large circles at the up and below of this figure represent two levels of looking at a problematic situation. While things are seen within a larger whole in connection to other things in the circle above, things are taken apart into the separated elements in the circle below. The sequence of thinking in CT and ST can be perceived in two opposite directions (arrows number 1 and 2 in Figure 1).

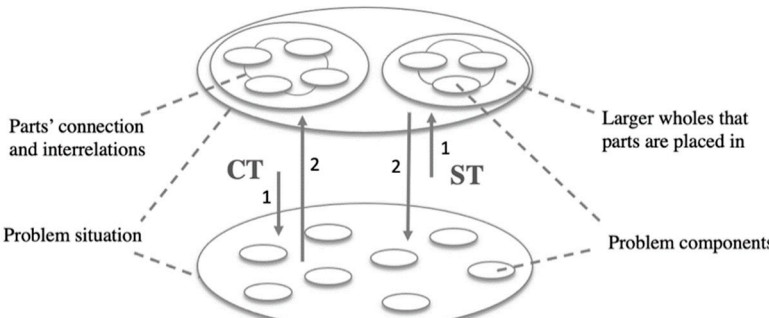

**Figure 1.** Thinking progress of CT and ST (arrows present the direction of thinking between two levels and numbers present the sequence of thinking).

The vertical navigation of a computational thinker in problem-solving is to take one step down into parts (that is problem decomposition practice of CT) and then take one step up to simplify and generalize the solutions to similar problems (that is abstraction and generalization concepts of CT) [6]. It leads to a reductionist way of thinking that articulates problem-solving by sorting out all parts of it [30]. Conversely, a systems thinker first navigates the way up to see a larger whole that parts are placed in (that is an ST concept known as holistic). Reductionism, the traditional "enemy" of ST, limits comprehension by requiring that a phenomenon be viewed from a simple and objective perspective [33]. Holistic thinking, which is at the core of ST, emphasizes emergence in a system, which is the behavior of the containing whole that is larger than the sum of its parts [30]. Algorithmic thinking, which is a fundamental concept of CT [4,6] if applied with the linear step-by-step procedure, leads to a straightforward casual problem-solving that is another source of reductionism [30]. Having said that, it is important to study if and how ST contributes to CT research theoretically and practically. Therefore, we elaborate on how ST bonds to CT in the literature in Section 5.

## 5. The Interplay of Computational Thinking and Systems Thinking

In this section we present some literature that applied an ST approach to CT research, including their research approach and contributions that exemplify the interplay of CT and ST. Looking at the literature related to CT, the interplay of ST and CT is diverse in the terms of referring to ST (either directly or indirectly), approach, and contribution (see Table 3).

**Table 3.** The interplay of CT and ST, research examples.

| Authorship | Reference to ST D: Directly I: Implicitly | Approach | Contributions |
|---|---|---|---|
| [13] | D | Supplementing CT with ST | • Supporting sustainability<br>• Considering the notion of change in complex problem-solving<br>• Improving critical thinking |
| [14,36] | D | Framing ST within CT practices | • Bridging the gap between theory and practice while integrating CT into STEM subjects |
| [3] | I | Applying the perspective-taking (rule) of ST to CT | • Introduction to CT from different perspectives |
| [37] | D and I | Applying perspective-taking and boundary setting (rules) of ST to CT | • Characterizing CT from different perspectives: CS-oriented, not CS-oriented, research-oriented |
| [12,38] | I | Applying boundary setting (rule) of ST to CT definition | • Underlying the ontological grounds of CT to consider the nature of it |
| [1,39–41] | I | Framing CT based on a holistic view of ST | • Developing CT theory dialogue<br>• Introducing new broaden features of CT<br>• Introducing a new version of CT |
| [15,42–45] | D | CT-ST integration | • Introducing a framework for developing CT and ST together |

The International Society for Technology in Education (ISTE) proposed an operational definition of CT emphasizing essential CT dispositions and attitudes for dealing with wicked problems and complexity [46]. Similar studies have been done by looking at challenges and persistence in the context of acquiring CT [35,47]. They include different skills to manage open-ended problems, not-transparent problems, and difficult ones in terms of complexity. Although it is not a direct quotation to ST, these characters are at the core of ST's perspective when facing complex issues.

The attempt to introduce ST to CT was developed by Weintrop et al. [14] and Ho et al. [36]. According to them, literate citizens with digital competence and CT skills should improve their ability to think systemically. They considered the importance of ST for CT development to bridge the gap between theory and practice, regarding the fact that CT is not taking place in a vacuum. While working on integrating CT into STEM education, the authors proposed a taxonomy of CT practices applicable to mathematics and science subjects. The taxonomy includes four categories that are data practices, simulation practices, problem-solving practices, and ST practices.

CT development has been extensively examined through six perspectives in the book titled "Computational Thinking" by Denning and Tedre [3]. Although ST is not addressed in their book, they have applied two ST principles (perspective and relationship) to define CT. Perspective-taking is one of the fundamental rules of ST approaches [48]. Each perspective in their book outlines a dimension of CT, but also goes hand in hand with other perspectives. It represents another rule of the ST approach—relationship— which is reflecting how one perspective is related to another one. These perspectives include methods, machines, computing education, software engineering, design, and computational science. From a methods perspective, CT is considered as a new way of

putting computing and reasoning to the work even by non-experts. From the perspective of the machine, they looked at the CT potentials that influence the evolution of computing machines. The other perspective, computing education, explores educational aspects of CT in connection to different fields. The software engineering perspective looks at CT-related activities to build software systems to overcome the unreliability of large software systems. From the design perspective, the notion of design CT is studied by addressing users' concerns and interests. The last perspective, computational science, is about CT and conducting scientific practices [3]. This latest perspective is the point of departure from CS, where CT is studied in other fields. It is noted that CS and *computational science* are different fields; meaning that where problem-solving through designing algorithms is taken-for-granted in CS, modeling and simulation for describing the behavior of a phenomenon are central in *computational science*.

The same authors introduced a new version of computational thinking (CT2.0) that represents a shift from rule-driven CT to data-driven CT [41]. Considering that data-driven approaches (for example, machine learning) have become commonplace in digital apps, tools, and services, there is a need to reconsider the conceptual landscape, educational practices, and technological methodologies of traditional rule-based CT1.0 related to teaching CT in K-12. This view outlines central concerns in traditional CT such as deductive reasoning, reductionism, and determinism that are covered with data-driven CT2.0. Thus, CT2.0 implies inductive reasoning in problem-solving that is emergent and strongly context-dependent [41]. Emergence and context (boundary) are central concepts of ST in tackling problems [30]. Data-driven CT has been touched upon very recently by Mike et al. [49], referring to *data thinking* as a new thinking skill that integrates CT into different domains. Having said that, the new version of CT relies on some fundamental elements of ST, though it is not labeled explicitly in CT2.0.

Underlining perspective-taking as the key element of ST, Xu and Zhang [37] introduced CT as the synergy of three perspectives: from outside of CS, from inside of CS, and from a research viewpoint. Not only have they applied ST as a general approach for characterizing CT, but they also moved further and introduced ST within the second perspective (from inside of CS) by addressing the role of ST to make CT more practical. Their approach implicitly comprises two key dimensions of ST including perspective-taking and boundary judgment.

DiSessa, in his studies, brought the idea of the big picture of CT [38]. To his view, CT is a social movement including good and not-good aspects. Looking at CT from this perspective underlines the ontological grounds of CT to consider its nature. He emphasizes that this view addresses two main issues related to CT including the cultural properties of CT as a movement, and the intellectual heritage of it. To our view, his standpoint is a systems approach since he has broadened the boundary under which the CT is examined.

In another study, while ST is not originally included in a prominent paper by Kafai et al. [39] on developing CT theory dialogue, we argue that their study is an ST view for framing different approaches to CT research. In their view, promoting CT in K-12 education is classified into three framings having a mutual dialogue with each other: cognitive, situated, and critical CT. The first framing—cognitive CT—expands on building CT skills for individuals, which is the dominant approach in CT research. Accordingly, CT concepts and practices will be formed and improved through engagement in CT training programs. The second framing is named situated CT. Drawing from the constructionist theory introduced by Papert [18], personal meaning and creative expressions will be developed when children use shareable digital artifacts in peer-supported CT activities. Engaging in situated CT programs supports students' social interaction and identity development. The third framing—critical CT—has developed more recently emphasizing existing structures of power, privilege, and agency at the society-at-large level. Social justice issues, political, and ethical challenges are the focus of CT applications within the third described framing that helps students to become computational literate [1].

As shown in Figure 2 below, moving from cognitive CT to critical CT embraces more holistic perspectives where CT is advancing to promote thriving, consciousness, and activism in understanding and enacting social changes. With the shift from cognitive to critical CT, one can consider that an ST approach is underlined. In the same way that a systems thinker is looking for pushing the boundaries to consider the impacts of the computer on marginalized people and society [33], a critical CT approach seeks to improve children's pragmatical, sociocultural, and political understandings to face real-world applications of computing [1,39]. Moreover, where in cognitive CT, skill development is practiced at the individual level based on CS concepts like algorithm and loop, people's roles and perspectives are the focus of situated and critical CT, particularly the latter.

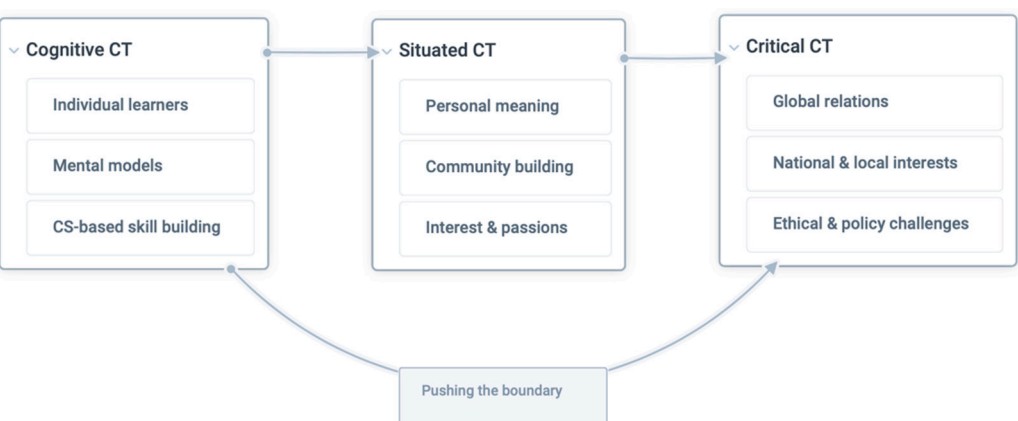

**Figure 2.** CT framings shift (adapted from Kafai et al. [39]).

More freshly, in line with the CT model mentioned above [39], a new framework has been proposed by Mills et al. [40] known as "Inclusive Computational Thinking". The framework consists of three concentric circles including CT skills, CT practices, and CT inclusive pedagogies placed at the core of the framework (see Figure 3). With a more holistic view, the innermost circle emphasizes newer aspects such as students' level of engagement, making a connection to their interests, community building, and taking a stand against inequity. These features address diverse interacting elements within a broader boundary at different levels that represent characteristics of ST highlighted by researchers [50].

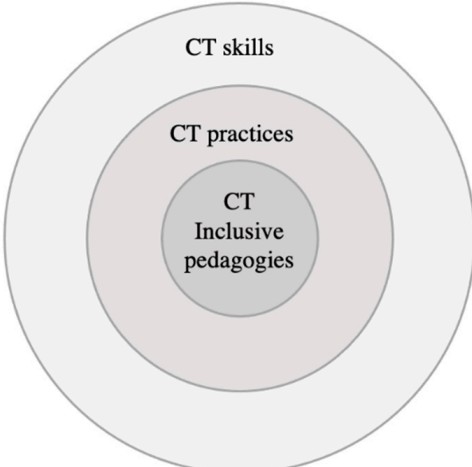

**Figure 3.** Inclusive CT framework (adapted from Mills et al. [20]).

The interplay of CT and ST is not always one-sided like using ST approaches to CT research. That said, there is a two-way interplay between CT and ST [15,42,43]. While from one side, ST approaches help students to develop their CT skills, having CT competence

contributes to understanding and responding to systems under study. It has been argued that modeling is a means for integrating CT and ST. As an example, multilevel system modeling is demonstrated in a study by Bowers et al. [15] as the link connecting CT and ST. To be specific, CT and ST are combined and contextualized when students engage in system modeling practices. For example, when students work on system modeling, they apply modeling practices from ST and testing/debugging practices from CT. In the same way, students define the boundaries of the problem and use the feedback loops from ST (see Table 2) and apply abstractions for representing data and problem-solving from CT. Another example is the theoretical framework proposed by Shin et al. [44], who manifested the aspects of CT and ST through computational modeling. They argued that CT and ST should be developed together to advance knowledge and skills for dealing with local and global problems. Study results conducted by Rachmatullah and Wiebe [45] reveal that while computer-based modeling activities in the context of food webs enhance CT and ST skills, there is a weak correlation between CT and ST skills. It means that the two constructs have a correlation, but they do not represent the same skills and are not located in the same cognitive dimension. This implies that the two constructs of CT and ST are distinct and separate, even though they may be related in some way.

The use of robotics in the classroom is considered to bring CT and ST together. The application of educational robotics in schools has been pursued by numerous researchers. Whereas robot programming fosters CT skills, the interaction of the robot with its dynamic surroundings through the use of sensors indicates ST development [22,23]. Therefore, different aspects of ST and CT are developed and interact simultaneously. Looking at the approaches and studies mentioned above illustrates that ST has gained attention either directly or indirectly in connection to CT research. We classified the ST approaches that are applied to CT in the discussion section. It helps us to find the gap in the knowledge and to make our research more focused.

## 6. Discussion

As mentioned in previous sections, the findings show a shortage of studies that explicitly bond ST to CT. However, looking at the few studies that include both ST and CT two primary connections have been grounded. In the first approach, ST is embedded in the CT practices. In the second approach, CT and ST are studied in parallel and CT is supplemented by ST. Yet, in a few studies the combination of both approaches can be outlined [51,52]. A complementary approach is then suggested at the end of the discussion section, which applies an ST lens to CT research that is regarded as the scientific contribution of this paper.

### 6.1. The First Approach: ST Is Positioned within CT Practices

In the first approach, CT is considered to be broader compared to other thinking skills for efficient and effective problem-solving [5] and ST is considered as a critical component of CT practices. Accordingly, ST is viewed as an element of CT while mainly referred to as perspectives [6]. According to this view, three aspects related to perspectives are developed through a CT practice [24]. The first aspect is when a computational thinker uses computation not only for computer-based problem-solving, but also for developing designing skills and self-expressing abilities. The second aspect is the value of interacting and collaborative work in terms of creating with others and for others. The third aspect is the ability of questioning, which includes both interrogation and response to that through design. Scholars argued that considering ST as an element of CT practices empowers thinking in levels, particularly in the context of complex systems [14,36]. Figure 4 represents the position of ST that is noticed in the first approach.

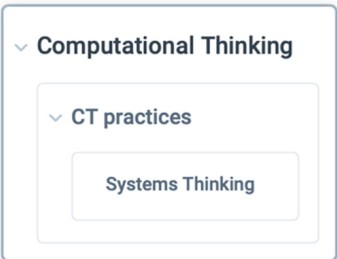

**Figure 4.** Positioning ST within CT practices.

Placing ST within CT as an element of CT practices encompasses the following purposes: studying a complex system as a whole, understanding the relationships within a system, thinking in levels from a micro-level view to macro-level view, communicating information about a system, defining the system in terms of its boundary, and managing the complexity [14]. Among these practices, we underline communicating information that is a challenge to sharing the knowledge and experience that one learns in practice. Collaboration and creativity are considered as elements of CT practices that are considered as part of twenty-first-century skills [6]. Improving collaboration, group work, and thinking out-of-the-box are examples of features associated with CT that are achievable through systems thinking. When ST is considered as part of CT practices in the educational context, it helps students to view the system that underlies the problem setting [53]. Moreover, it enables them to transfer the solution to other problems within the same system.

*6.2. The Second Approach: CT Is Supplemented with ST*

In the second approach, which is traceable in relatively new frameworks, CT is suggested to be supplemented with ST. This approach responds to social challenges and real complex problems that are a part of the problem-solving process, so-called "critical computational thinking" [39,54,55]. Critical CT refers to the interdependency between society and CT that develops children's awareness of matters behind technology, such as dynamic social issues [51]. It also helps deal with other issues and deficiencies revolving around CT, such as dogmatism, reductionism, technical solutionism, and a narrow view of computing connected to CT [13].

Along with the CT studies that directly referred to systems thinking as the complementing skill, findings show a few papers that embrace the systems thinking concepts and foundations supplementing the CT and vice versa. Research carried out by Berland and Wilensky [56] provides a better understanding of the connection of CT with complex systems thinking. The authors used the physical and virtual robotics platforms to operationalize the sense of CT in connection to ST. Complex systems thinking in their study reflects the levels of thinking in terms of components of the system, accumulation of the components, and emergent meanings out of their relationship. The result of their study illustrates that CT and complex systems thinking are mutually reinforcing. In connection to STEM subjects, Holme [57] discussed that ST is a vehicle to enhance CT skills in general chemistry. Other research conducted by Pancratz and Diethelm [52] focuses on developing part–whole thinking skills for applying CT in educational settings. Part–whole thinking, which contains the aspects of systems thinking, plays an essential role in the context of CT, according to their findings.

It is emphasized that ST and CT are supplementing each other through computational modeling [15,42,43]. Eidin et al. [42] argued that CT and ST skills are together manifested through computational modeling practices. According to their view, in the process of constructing the computational models, central skills related to CT and ST would be improved. That is, boundary judgment of the system, designing and constructing, testing and iterative refinement, and explaining and predicting the behavior of a system. The same idea has been applied by Haas et al. [43], investigating how computational modeling

promotes the ST skill of students. Their results suggest that computational modeling, as one component of CT, enriches the context to foster ST among students.

Together with supplementing CT with ST from a general level, the constructs of CT (concepts and practices) could also be augmented with an ST perspective (see Figure 5). For instance, logical thinking, which is considered a primary construct of CT and builds analytical thinking skills [6], can be supplemented with an ST perspective.

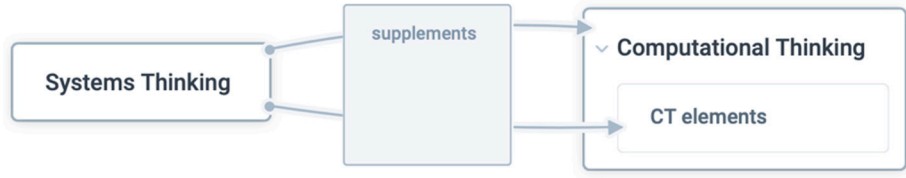

**Figure 5.** Positioning ST parallel to CT.

To better make sense of it, Boolean operators such as IF, AND, OR, and NOT are examples of applying logic in decision-making. Considering that mentioned logics used in CS and CT are bi-valent, which take two discerning positions (that is right or wrong, true or false), they do not look at the gray-shade spectrum between the right and wrong values that most likely exist in reality. According to Cabrera and Cabrera [32], a unifying theory of ST provides the route to take multivalent logics in more complex, messy, and wicked situations. Accordingly, bivalent logic in real-world problems is a part of the multivalent system of logic. Thus, multivalent logic implies the existence of other aspects of ST such as perspectives and relationships among elements of a system.

*6.3. A Complementary Approach: Looking at CT through the Lens of ST*

In this subsection we will try to explain and elaborate on what a complementary approach is for studying and analyzing CT. As previously indicated in Section 4, traditionally CT has been seen and developed as a cognitive and information processing point of view for problem-solving. This viewpoint implies a reductionist approach that disregards the varying perspectives relating to the problem in question. The idea of bonding CT to ST was proposed in 2014 by Easterbrook [13]. It is highly cited thereafter; however, there have been few studies focusing on the interplay of CT and ST since then [57]. We classified existing approaches of CT research in connection to ST in the previous subsections. Beyond them we propose a complementary perspective, which is looking at CT through the lens of ST. Using ST as a lens to look at CT is a broader perspective that provides an additional complementary tool to CT research. To our knowledge, this approach is lacking in the existing literature.

In line with the idea of the big picture of CT provided by DiSessa [38], we would look broadly into multiple niches and corners of CT in education to see where its position is now and where it is heading to. According to Reynolds and Holwell [31], ST reveals the underlying features of the situation. Considering the many aspects of CT integration into the educational context regarding the policies, methods, programs, perspectives, boundaries, stakeholders, age levels, and evaluation, it is essential to look at the whole situation with a broader view through an ST lens. Conflicting positions in CT literature in terms of definitions, concepts, practices, techniques, tools, strategies, target groups, and evaluation make it much more complex to conceptualize a CT framework that can be applied to different fields [58]. From another perspective, ST can be used as a common language across different methods, techniques, approaches, and contexts [48] related to CT. It helps to draw the boundaries either conceptually or practically, and to clarify the relationships between different factors involved in the system. Putting a boundary around something, and in our case around CT, affects how we research. It not only gives a new identity to it, but deliberately separates the CT aspects that should be studied and the methodological approaches that could be used [30,33,48]. At the same time, it would

identify the part–whole characteristics involved in CT by considering as many perspectives as possible.

Figure 6 depicts a potential application of the ST lens to CT research where the boundary critique [59] has been applied from the perspective of stakeholders who are involved in CT development in schools. Boundary critique is a method in critical systems thinking, which is used to analyze the boundaries of a system and its interactions with other systems and to question the underlying assumptions, values, and power dynamics that shape the system. As seen in this picture, it enables us to shift from the initial boundary of the system to a proposed second boundary.

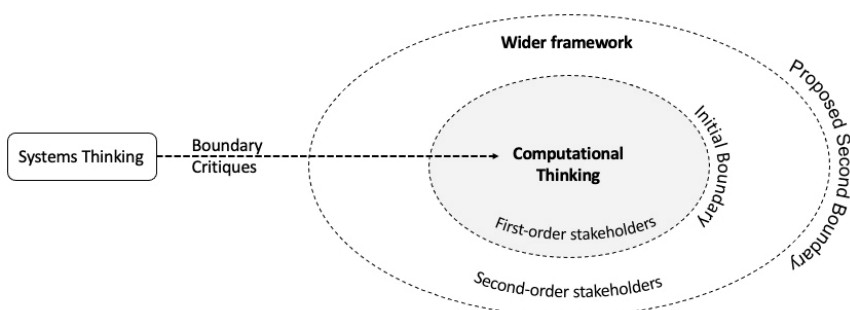

**Figure 6.** Looking at CT through the lens of ST, a boundary critique approach.

First-order stakeholders within the initial boundary include those who are directly influenced when CT is integrated into school subjects, such as in-service teachers and students. Second-order stakeholders positioned in the proposed second boundary are those who are indirectly affected by the system, such as school principals and pre-service teachers. When studying a system, it is important to consider both first-order and second-order stakeholders, as their perspectives and concerns can be very different and can impact the system in different ways. Understanding the needs and interests of both types of stakeholders can help to ensure that a system is designed to meet the needs of all those affected by it.

## 7. Conclusions

CT is a fundamental skill that is the focus of many disciplines to improve the problem-solving abilities of individuals in today's computing-embedded society. The current paper applies an IS lens to CT by deploying ST as an accredited theory in the IS field. ST approaches are considered as a link to bringing CT into IS field. Incorporating ST into CT research broadens the focus from technical aspects to include the societal and behavioral impacts on those involved, such as students, parents, teachers, administrators, and policymakers. On one side, CT cannot develop or function independently of other fields, such as Information Systems in this context. Therefore, integration of CT into IS research would be beneficial for students to develop problem-solving skills by looking beyond the reductionist nature of CT and for educators to have access to more resources. On the other side, to develop an IS discourse, theorizing from other disciplines and synthesizing them for contributing to IS field should be taken into consideration.

The interplay of CT and ST is relatively new in the literature responding to the flaws and risks associated with CT. Problematizing CT with an ST lens looks promising to push the boundaries of CT research toward developing societal and behavioral aspects of CT for dealing with real-world complex problems. Among different studies which either explicitly or implicitly presented the interplay of ST and CT, we recognized two main approaches where ST is used in CT research. ST is embedded in the CT practices in the first approach, and ST is considered a complementary view to deal with social deficiencies of CT research in the second approach. In the second approach, not only CT is supplemented by ST, but both would also be improved while they are simultaneously used in a given context. Accordingly, the complementary approach we propose is to look at CT from

the ST perspective to provide a clearer picture of the concept for the long-term strategic development of CT in the educational context and to broaden the involved scenarios in CT development.

That said, we encourage CT researchers to consider the following implications of our work:

(1) to address issues emerging from the reductionist nature of CT for problem-solving and reflect on its multifaceted attributes;

(2) to address issues in broadening contribution in CT research from other fields that could be encouraged by inclusive discourse and practices of what CT is, who engages in CT research and practice, and who is capable of shaping that discourse;

(3) to build theoretical contributions to the CT literature.

This complementary view is the outcome of our initial efforts that could help improve traditional CT research with an ST perspective. As a next step, we need to further evaluate these ideas by developing a framework that will be validated with empirical data in a number of experiments. We orient our ongoing activities by engaging different elements for developing CT in K-12 education. From a practical point of view, our efforts are focused on the application of educational robotics in different contexts (formal and informal) by taking perspectives from pre-service teachers, in-service teachers, students, and researchers [23,60,61]. From an ST perspective, we will start to explore the interconnections between involved factors to form a systemic view of CT development in K-12 education.

We aspire to help CT researchers in attempting to better apply theories from other fields, in order to broaden the factors impacting CT research and enhance learning. This will inform the work we plan to carry out in the near future.

**Author Contributions:** Conceptualization, A.H.; methodology, A.H.; resources, A.H.; writing—original draft preparation, A.H.; writing—review and editing, A.M. and M.M.; supervision, A.M. and M.M.; All authors have read and agreed to the published version of the manuscript.

**Funding:** This research is funded by the Swedish Research School of Management and IT (https://www.mit.uu.se/?languageId=1, accessed on 15 September 2022) together with department of Informatics, Linnaeus University, Växjö, Sweden.

**Data Availability Statement:** Data sharing is not applicable to this article.

**Conflicts of Interest:** The authors declare no conflict of interest.

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
