# Peer review of "A Complementary View to Computational Thinking and Its Interplay with Systems Thinking"

_education, doi:10.3390/educsci13020201_

Round 1
Reviewer 1 Report
The paper proposes different approaches for connecting Computational Thinking with Systems Thinking. It explores current research in this regard in depth, finding essentially two approaches to this connection and proposes a third.
The paper is well written and is not superficial. Moreover, the proposed approach is very interesting, as it could alleviate some of the operational problems we still encounter in applying CT in classrooms. Below are some comments and suggestions to the authors that I think could improve their paper:
- I recommend a complete revision of the text, as there are some grammatical and syntax errors, for example:
o Paragraph 2, first paragraph: "In fact it focus ..." instead of "In fact he focus ..."
o Paragraph 2, third paragraph: "Brennan" instead of "Brennen".
o Paragraph 5, last paragraph: "...our research..." instead of "...my research..." o Paragraph 5, last paragraph: "...our research..." instead of "...my research...".
o Paragraph 6, third paragraph: In the list, some items are not numbered, it ends at number 4.
- When talking about computational thinking in section 2, it is very important to distinguish clearly between (1) generic definitions (e.g. Wing's definition, (2) decompositions of CT into facets (e.g. abstraction, decomposition, data analysis, debugging, algorithmics, generalisation, ...) and (3) operational definitions for the application of computational thinking in the classroom, including Brennan and Resnick's three-dimensional framework and Grover and Pea's redefinition of it. This differentiation is not made clear in the introduction. In addition, it should be explained that Brennan and Resnick's framework is based on a student activity while creating a video game with Scratch, which can be more easily linked to systems thinking.
- In this same section, it should also be noted that there is a lot of research on the use of robotics in the classroom for the development of computational thinking, as it is mentioned below as a suggestion, when there is a lot of research and case studies already documented.
- Again, in this same section 2, Brennan and Resnick's definition of the computational thinking practices dimension is shown as follows: "... or how CT improves?", this is not the original definition provided by these authors and I suggest that it be modified, as it is not accurate.
- In the last paragraph of section 2, it is mentioned that we are talking about computational thinking for beginners in primary education and it is commented that computational thinking for professionals includes other aspects such as virtual reality. This is not always the case, but only a proposal of certain researchers, since normally, the generic definition of computational thinking and its decomposition into facets is the same for all age groups, and if these other aspects are included, it is also done from early stages and not only for adults.
- Finally, this section should also mention dispositions and predispositions to develop computational thinking, which include other aspects such as persistence or frustration tolerance (possibly these can also be connected to IS).
- In section 4, it could be included that many authors also consider the social aspect of computational thinking, in fact, Brennan and Resnick's computational perspectives dimension is based on this social dimension.
- This section 4 rightly talks about the concepts ((2) decompositions or facets) of CT, but there is no connection to this in section 2 (as mentioned above). I think that making such differentiations would make everything clearer.
- Congratulations on Table 3, it is very clarifying.
- In section 5, second paragraph, here dispositions and attitudes are mentioned for the first time, when they should already have been mentioned in section 2 (as mentioned above), e.g. the author R. Israel-Fishelson, already mentions these kinds of attitudes in quite some depth.
- In the paragraph after Figure 3, an example of research using robotics in the classroom could be included (I recommend e.g. author Nardie Fanchamps).
- The discussion section is very good, only, a little more systematic analysis is missing in the last approach (section 6.3), as it is the approach proposed by the authors which is very interesting and promising, but it is too superficially described. Perhaps it would help if Figure 6 were worked on a bit more to make it more descriptive, including some of the most important ideas in it.
- Finally, in the conclusions, digital competence is described as a broader framework than computational thinking. In my opinion this is not the case at all, as they are very different areas, one is not included in the other, even if they have some aspects in common.
- In the conclusions it is mentioned that researchers will focus on applying robotics in the classroom. As I said before, I think it is a promising approach to establish the vision of computational thinking from IS, but it should be reflected in the paper that there is already a lot of research on the application of robotics in the classroom for the development of CT.
Overall it is a good reflection and a different and promising approach to CT.
Reviewer 2 Report
The manuscript was reviewed. My report is given below:
1- Why was such a study needed? What gap in the literature will the study fill? It should be indicated by a literature review.
2- CT and ST are the subjects of intensive study. The literature can be expanded to be up to date.
3- The problem of the research should be revealed and the text should be structured on this problem.
4- IT concepts can be taken in more detail in Table 1. For example data representation (data collection, analysis, presentation), modelling.. Why were these concepts not taken?
5-The discussion should be written on the research problem.
6- Suggestions should be made directly related to the results of the research.
Round 2
Reviewer 2 Report
It is seen that the article has been revised in line with the recommendations of the referees. I think this is acceptable as it is.